# The Lean Six Sigma Define, Measure, Analyze, Implement, Control (LSS DMAIC) Framework: An Innovative Strategy for Quality Improvement of Pharmacist Vaccine Recommendations in Community Pharmacy

**DOI:** 10.3390/pharmacy10030049

**Published:** 2022-04-22

**Authors:** Kenneth C. Hohmeier, Chelsea Renfro, Benjamin Loomis, Connor E. Alexander, Urvi Patel, Matthew Cheramie, Alina Cernasev, Tracy Hagemann, Chi-Yang Chiu, Marie A. Chisholm-Burns, Justin D. Gatwood

**Affiliations:** 1Department of Clinical Pharmacy and Translational Science, College of Pharmacy, University of Tennessee Health Science Center, Nashville, TN 37211, USA; upatel9@uthsc.edu (U.P.); acernase@uthsc.edu (A.C.); thageman@uthsc.edu (T.H.); jgatwood@uthsc.edu (J.D.G.); 2Department of Clinical Pharmacy and Translational Science, College of Pharmacy, University of Tennessee Health Science Center, Memphis, TN 38163, USA; crenfro@uthsc.edu (C.R.); calexa34@uthsc.edu (C.E.A.); mcherami@uthsc.edu (M.C.); chiu@uthsc.edu (C.-Y.C.); mchisho3@uthsc.edu (M.A.C.-B.); 3Walgreen Co., Nashville, TN 37214, USA; benjamin.loomis@walgreens.com

**Keywords:** Lean Six Sigma, community pharmacy, DMAIC, vaccination, pneumonia vaccine, quality improvement

## Abstract

Community pharmacies represent a highly accessible and convenient setting for vaccination. However, setting-specific barriers exist which contribute to suboptimal vaccination rates, particularly for pneumococcal vaccinations. One proven quality improvement framework growing in use within healthcare settings is Lean Six Sigma (LSS). This paper describes the application of the LSS framework in select locations of a national pharmacy chain. The implementation of a training program for improved recommendation techniques to promote higher rates of pneumococcal vaccinations in high-risk adult populations is also addressed. A mixed-methods approach including pre/post quasi-experimental design and in-depth key informant interviews was used.

## 1. Introduction

Vaccinations are key interventions to help in protecting against and significantly reducing the risk of many diseases in patients. In particular, pneumococcal vaccinations provide protection for patients aged 65 and older and those at high risk for acquiring the disease. The two pneumococcal vaccinations available in the United States are the pneumococcal conjugate vaccine (PCV13) and pneumococcal polysaccharide vaccine (PPSV23) [1]. Current recommendations for pneumococcal vaccinations provided by Centers for Disease Control and Prevention include the PPSV23 in high-risk adults from 18 to 64 years of age and for all adults 65 and older, and the PCV13 in certain high-risk populations in the United States [1]. Studies relating to PPSV23 and PCV13 efficacy demonstrate high estimates of clinical effectiveness for both vaccines [1]. Overall, the PPSV23 vaccine has shown approximately 70% clinical effectiveness and PCV13 has shown 75% clinical effectiveness in preventing invasive disease caused by the bacterial serotypes [1].

Even with such high levels of efficacy of these vaccines, pneumococcal vaccination rates in the U.S. are currently below the desired benchmarks to improve health outcomes [2]. A number of factors contribute to such suboptimal vaccination rates with vaccine hesitancy being a top priority on the list. The World Health Organization listed vaccine hesitancy as one of the top 10 threats to global health outcomes in 2019 [3]. Vaccine hesitancy has been defined as “a delay in vaccination or a refusal to vaccinate in spite of vaccine availability” [4]. The majority of vaccine hesitancy research has been conducted in the pediatric population; however, there has been a growing interest in evaluating vaccine hesitancy in the adult population, a field that is currently under researched [5].

As certified immunizers and patient health advocates, community pharmacists play a vital role in impacting vaccination rates. Pharmacist recommendations for vaccination are generally effective, but success varies widely, ranging from 50 to 90% of patients receiving the recommended vaccine [6]. Such wide ranges in effectiveness are a common target of healthcare continuous quality improvement (CQI) efforts, including CQI using the Lean Six Sigma (LSS) framework [7]. Lean Six Sigma is a method that provides an organization processes and tools with the aim of either improving their performance, reducing process variation, or both.

The purpose of this paper is to describe the application of the LSS framework within a national pharmacy chain to improve vaccination rates and reduce variability in vaccination recommendation acceptance across sites. The implementation of an evidence-based training program for improved vaccine recommendation technique is also discussed.

## 2. Materials and Methods

### 2.1. Lean Six Sigma

LSS and DMAIC (define, measure, analyze, improve, control) methods were selected for this CQI project given their increasing use in healthcare, especially as it relates to reducing process variation, a significant concern of the authors regarding vaccination recommendation processes.

In general, LSS methods have become increasingly integrated across a variety of healthcare settings over the past two decades [7,8,9,10]. A recent study reported the use of LSS methods in reducing pharmacy wait times [9]. LLS methods in this study were used to identify potential problems associated with increased wait times and found that implementing technology, such as automated queuing, pharmacy devices for quick and accurate filling and dispensing, and computer simulation modeling for smooth workflow, were solutions to the issue. DMAIC is a specific quality improvement strategy within LSS that includes defining a problem (D), measuring the problem (M), analyzing the cause of the problem (A), implementing a feasible solution (I), and controlling the improved process to ensure maintenance of any gains in capabilities (C). A 2021 study also used LSS DMAIC to reduce variations within medication synchronization workflow and reduced packaging time within workflow by ~70% [10].

### 2.2. Practice Site

The LSS DMAIC initiative was conducted across two divisions of a national pharmacy chain in the Southeast United States (Walgreens Co., Deefield, IL, USA). Most sites were located within the metropolitan areas of Memphis, TN or Nashville, TN. Sites were selected in collaboration with pharmacy chain leadership. All pharmacies in the Nashville (*n* = 46) and Memphis (*n* = 50) regions were included.

### 2.3. Evaluation

A mixed-methods approach including pre/post quasi-experimental design and in-depth key informant interviews was used in analyzing the LSS intervention. Interviews were conducted by trained student research assistants over the telephone, recorded digitally, and subsequently transcribed verbatim by a third-party transcription service. Quantitative data analysis included both descriptive and inferential statistics, including Mann–Whitney U tests, chi-square tests, and general linear models. An alpha level of 0.05 was selected. Qualitative data analysis included content analysis performed by two researchers trained in qualitative methodology (KCH, AC). A phenomenological approach was used. The project was reviewed and deemed exempt by the University of Tennessee Health Science Center (UTHSC) Institutional Review Board (IRB).

## 3. Case Study

The implementation of the LSS DMAIC is described across each of the phases of process.

### 3.1. Define

The problem of interest was variations in vaccine acceptance rates across pharmacy sites. A project charter was developed by the project team to define the scope, goals, and timeline of the project. Voice of the customer (VoC) data was derived from a scoping literature review and internal customer service data. A goal of a 20% increase in pneumococcal vaccines delivered was set as the new specification based on conversations between the project team and chain pharmacy leadership.

### 3.2. Measure

Baseline (i.e., “current state”) rates of pneumococcal vaccinations administered at each pharmacy were compared across the pharmacy division. Pharmacy site demographics were abstracted from the pharmacy chain organization to aid in analysis. A high-level map of vaccine recommendation processes, as shown in Figure 1, was developed in collaboration with chain pharmacy division leadership.

### 3.3. Analyze

Baseline data from historical pharmacy dispensing data was used to understand correlations between staffing and hours of operations and vaccination rates, but no significant correlations were identified after regression analysis. A cause and effect matrix along with a failure modes and effects analysis (FMEA) were conducted as shown in Table 1 and Table 2, respectively. The FMEA was used to identify all potential failures possible in the vaccine recommendation process. As a result, the vaccine recommendation communication technique used by the pharmacist was selected as the root cause hypothesis. A high-quality pharmacist-provided recommendation was deemed to overcome the 5 of 6 failure modes.

### 3.4. Improve

A training program for pharmacists on quality vaccination recommendations was selected as the intervention to address vaccination recommendation process performance. The training was directed towards study site pharmacists and included both online and live training sessions [11]. The training program centered on the concept of “presumptive” vaccine recommendations [12], motivational interviewing [13], and the transtheoretical model [14] to guide pharmacists through behavior change toward vaccine recommendation acceptance. The live sessions were led by faculty facilitators. Pharmacists completed the training program in July 2019.

A design of experiments (DOE) process was used to isolate variables, including vaccination rates, time, pharmacist experience, and patient hesitancy. Key informant interviews provided context on processes in community pharmacy workflow. Content analysis of interview transcriptions highlighted vaccination hesitancy and related barriers present during pharmacist recommendations.

A capability analysis test was conducted to evaluate the process capability, Cp, Cpk, and *p*-value. In total, 25 pharmacists completed the online and live components of the vaccine recommendation training program. There was a 24% difference in vaccination counts when comparing sites that underwent the full training versus those that did not (*p* < 0.05). A total of 54 pharmacies completed the experiment, 25 intervention pharmacies and 29 control pharmacies. Of the 25 intervention pharmacies, 8 (32%) increased by the specification threshold goal of 20%. Of the 29 control sites, only 4 (13.7%) increased by the specification of 20%.

Thematic analysis of key informant interviews from intervention pharmacists uncovered four themes: (1) knowledge of importance of a presumptive recommendation wording, (2) trialability of a presumptive recommendation approach facilitates implementation, (3) changes in outcome expectancy facilitate presumptive recommendation approach implementation, and (4) barriers still present.

Pharmacists noted that selecting the appropriate wording to communicate the vaccination recommendation was a driver of recommendation acceptance. This theme related to the practical use of the training within the pharmacy itself. Online and simulation training emphasized the importance of presumptive vaccination language, and pharmacists’ individual experiences using this approach solidified the effectiveness of this approach.


*“So I think the, like what we learned, the main thing was just the verbiage. Instead of just asking the patients, “do you want to get a [vaccination]?” Just saying, “you’re due for your [vaccination]. Which arm do you want to get it in today?” So I think that was the main thing is just our verbiage with it.”*
(R1)

Pharmacists made note of the low stakes “trialability,” that is the initial, experimental use of the training, increased uptake. A relatively low amount of effort was required on the part of the pharmacist to implement vaccination recommendation training. Primarily, only a “psychological” commitment was required for engaging in the use of presumptive vaccination recommendation language—as vaccine recommendations were already a component of workflow, required as part of job duties of the pharmacist, tracked via internal pharmacy metrics, and socially accepted by patients, pharmacy technicians, and pharmacists. For this reason, pharmacists who were trained and those who staffed at the pharmacies where trained pharmacists worked indicated positive experimentation with the presumptive recommendation approach.


*“So I talked to our other pharmacists and asked them, I was like, “do you normally get people to say, yes, when you ask them if they’ve done their flu shot?” And, of course, he said, no. So I told him the other way we learned instead of saying, “have you gotten your flu shot,” is, “you’re due for your flu shot.” And so he just tried it on a couple of patients, and it worked. And so that worked well.”*
(R3)


*“I think we are at our goal for [vaccinations], which I wasn’t at this store last year, but I’m hearing from the store manager and the pharmacy manager that’s really, really good because a lot of the stores in our district are not at their goal yet. And I really think that’s because of the [vaccination recommendation training program] that I had and just bringing back as much of that and putting that in the pharmacy as I could.”*
(R7)

Pharmacists noted that pneumococcal vaccination recommendations were generally poorly received by patients, which in turn led to pharmacists’ hesitation to make the recommendation. Key informant pharmacists noted that the use of the presumptive recommendation language led to a change in outcome expectancy from a negative outcome to a positive one.


*“Well, overall, it makes it more comfortable for me to go talk to someone about it. And, if I have, you know, gotten more people vaccinated, then they benefit.”*
(R5)

Despite generally positive comments about the impact of the presumptive recommendation approach and its feasibility and effectiveness in real world practice, pharmacists noted that barriers of time still exist and this may hinder overall vaccination recommendation acceptance.


*“Just when we’re really busy, sometimes it’s hard to have time to step aside and [make vaccine recommendations].”*
(R2)

There was found to be overall convergence between quantitative and qualitative data on the vaccine training program’s positive impact on the pneumococcal vaccination process. In particular, the presumptive vaccine recommendation language was a key driver of success as it was a low cost, feasible, and effective alternative to traditional vaccine recommendation language.

### 3.5. Control

A quality control program was initiated after the training program by the pharmacy chain. Goals and metrics were assigned to pharmacies based on program results. Pharmacists were also provided guides of the vaccine recommendation process, as represented in Figure 2, to attach to their computer monitors at their respective sites to remind and ensure that each eligible vaccine recommendation is strong and assertive.

## 4. Discussion and Conclusions

LSS and DMAIC quality improvement frameworks represent an important tool for healthcare process improvement. In this study, we presented a case study of how DMAIC may be used to address both the needs of the organization (i.e., a community pharmacy) and public health at-large.

Importantly, in many organizational improvement efforts there is a bias toward “prescribing before diagnosing,” whereby pharmacy stakeholders solely use past experiences as a means to determine what solution should be implemented to address a problem of importance to them. DMAIC requires that stakeholders first take an objective viewpoint of the situation, defining problems in the context of the patient/customer (i.e., voice of the customer) and then systematically assessing a process to understand where deviations from the “goal” may exist.

In the present paper, we took this approach to better understand the vaccine recommendation process and where there were defects within that process. Although often cited barriers to clinical service delivery include things such as time, personnel, and technology [15,16], these “diagnoses” are often made subjectively and without the guidance of a structured CQI process such as LSS. Although time, personnel, and technology may have facilitated increased pneumococcal vaccination recommendation uptake, our objective analysis using DMAIC found that it was addressing the delivery of the recommendation itself that was the lowest cost, highest yield intervention that could be made. Our mixed-methods analysis then confirmed this hypothesis.

There were several limitations to our study. This study was conducted in a single region of a large U.S. pharmacy chain and this may limit generalizability. Furthermore, sites were not randomized for control or intervention, but rather pharmacy leadership selected sites based on organizational factors outside the control of the study team. Finally, this was a quasi-experimental pre/post design with a nonequivalent control group and thus secular bias may pose a risk.

Future directions for pharmacy stakeholders, including researchers, leaders, practitioners, and policy makers should include thorough contextual analysis of all possible barriers and facilitators for clinical pharmacy services, including vaccination programs, medication therapy management, point-of-care test-and-treat models, and others [17]. This contextual analysis, sometimes referred to as contextual inquiry, often uses a mixed-methods approach of quantitative and qualitative data analysis to understand all possible barriers and facilitators to achieving a given objective. On the pragmatic side of healthcare, this is often called CQI as in our study; however, on the research side of healthcare this is referred to as implementation science [17,18]. In either case, there is a plethora of tools, theories, frameworks, and methods available across both ends of this spectrum which may aid pharmacy stakeholders in improving the overall quality of the pharmacy experience [19].

## 5. Conclusions

Quality improvement frameworks, such as LSS DMAIC, may be used to both improve pharmacy operations as well as enhance patient care.

## Figures and Tables

**Figure 1 pharmacy-10-00049-f001:**
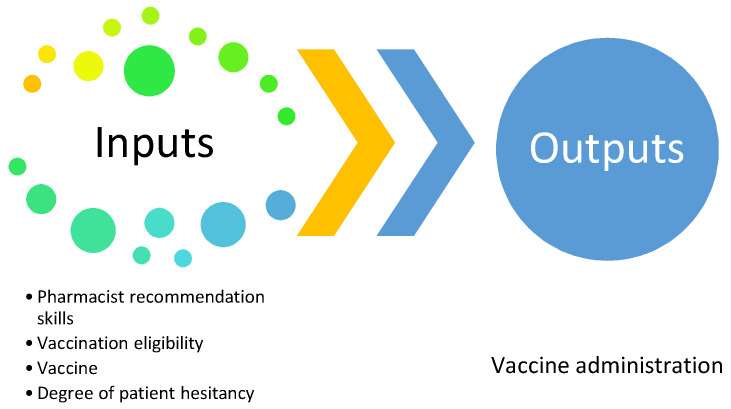
High level Lean Six Sigma DMAIC process map.

**Figure 2 pharmacy-10-00049-f002:**
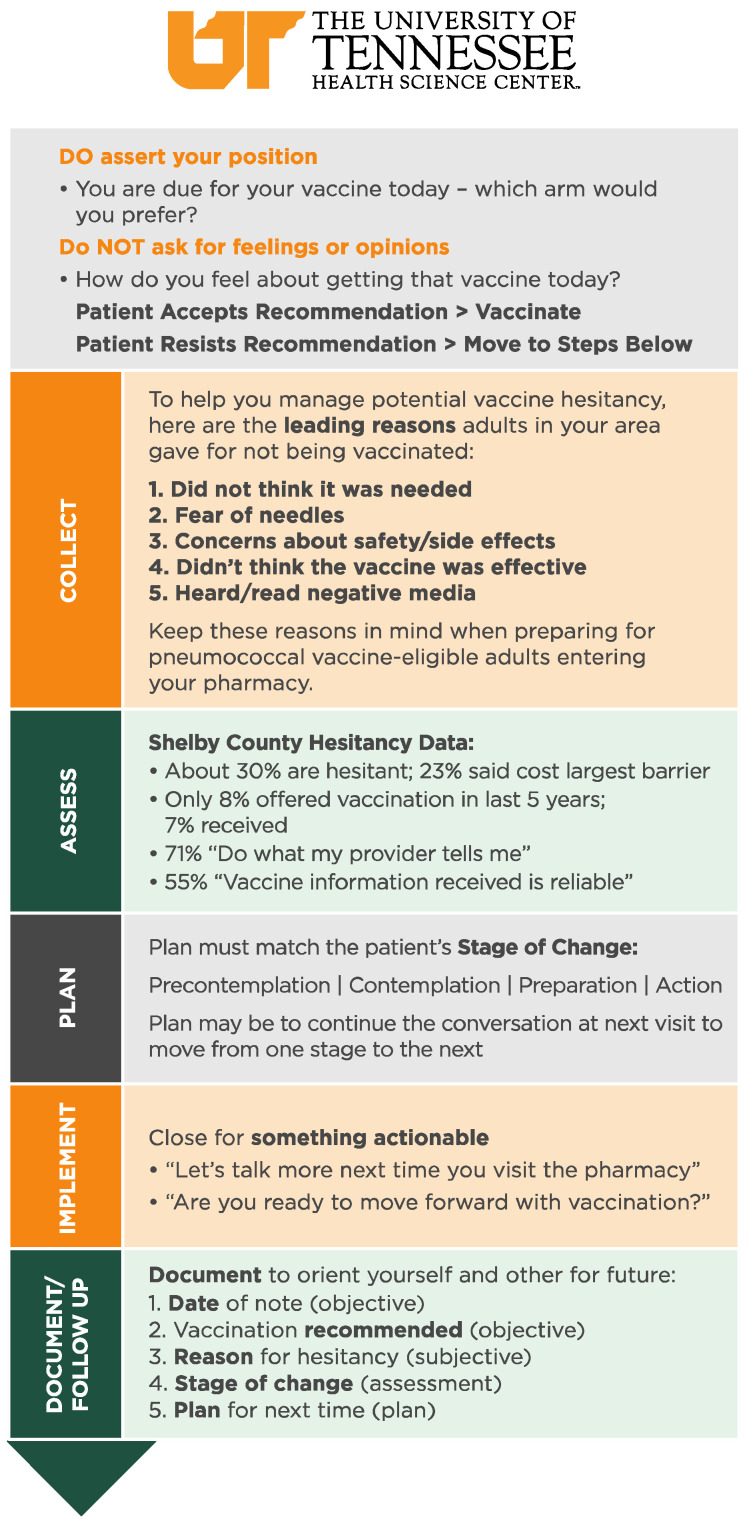
Presumptive vaccination recommendation pharmacy guide.

**Table 1 pharmacy-10-00049-t001:** Cause and effect matrix.

Rating of Importance to Customer →			10	
	Process Step	Process Inputs	Number of Vaccines	Total
1	Process Step 1	Initial Vaccine Eligibility	9	90
2	Process Step 1	Recommendation of Vaccine	9	90
3	Process Step 1	Vaccine Cost (hesitant patient)	6	60
4	Process Step 1	Patient Objective Information (hesitant patient)	9	90
5	Process Step 1	Patient Subjective Information (hesitant patient)	3	30
6	Process Step 2	Needle and Syringe	3	30
7	Process Step 2	Vaccine	9	90
8	Process Step 2	Technique/Training	9	90
9	Process Step 2	Pharmacist Experience	6	60
10	Process Step 2	Time (pharmacist)	9	90
11	Process Step 2	Time (patient)	9	90
12	Process Step 2	Informational Pamphlet	3	30
13	Process Step 3	Patient Education	3	30
14	Process Step 3	Counseling	3	30
15	Process Step 3	Vaccine Administration	9	90
Total				990

**Table 2 pharmacy-10-00049-t002:** Process/product failure modes and effects analysis.

Process Step	Key Process Input	Potential Failure Mode	Potential Failure Effects	SEV	Potential Causes	OCC	Current Controls	DET	RPN	Actions Recommended	Resp.	Actions Taken	SEV	OCC	DET	RPN
What is the Process Step?	What is the key Process Input?	In what ways does the key input go wrong?	What is the impact on the key output variables (customer requirements) or internal requirements?	How severe is the effect to the customer?	What causes the key input to go wrong?	How often does cause or FM occur?	What are the existing controls and procedures (inspection and test) that prevent either the cause or the Failure Mode? Should include an SOP number.	How well can you detect cause or FM?		What are the actions for reducing the occurrence of the cause, or improving detection? Should have actions only on high RPN’s or easy fixes.	Who is responsible for the recommended action?	What are the completed actions taken with the recalculated RPN? Be sure to include completion month/year.				
Determine vaccine eligibility	Initial eligibility	Patient is ineligible to receive vaccine	Patient cannot receive vaccine	9	Uncontrollable	1	Vaccine is only recommended to patients eligible to receive it	3	27	None	None	None				0
Lead with assertive recommendation	Patient subjective information (hesitant)	Recommendation increases patient’s hesitancy	Patient elects to not receive vaccine	9	Pharmacist is not properly trained	7	None	9	567	Pharmacists receive training on how to address vaccine hesitant patients	Pharmacist	Pharmacists receive virtual and in person training on how to handle hesitant patients (3/20)	9	3	3	81
Assess	Initial eligibility	Patient is ineligible to receive vaccine	Patient cannot receive vaccine	9	Uncontrollable	1	Vaccine is only recommended to patients eligible to receive it	3	27	None	None	None				0
Plan	Time	Patient does not have enough time to receive vaccine	Patient cannot receive vaccine	7	Pharmacy is understaffed	3	None	5	105	None	None	None				0
Implement (administer)	Pharmacist experience	Pharmacist does not know how to address hesitancy	Patient elects to not receive vaccine	9	Pharmacist is not properly trained	7	None	9	567	Pharmacists receive training on how to address vaccine hesitant patients	Pharmacists receive training on how to address vaccine hesitant patients	Pharmacists receive virtual and in person training on how to handle hesitant patients (3/20)	9	3	3	81
Follow-up/document	Patient education	Patient does not receive any follow-up or counseling	Patient is misinformed	5	Pharmacist is too busy	3	None	3	45	None	None	None				0
									0							0
									0							0
									0							0
									0							0
									0							0
									0							0

## Data Availability

Data may be made available upon request to the corresponding author.

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
