# Peer review of "The Lean Six Sigma Define, Measure, Analyze, Implement, Control (LSS DMAIC) Framework: An Innovative Strategy for Quality Improvement of Pharmacist Vaccine Recommendations in Community Pharmacy"

_pharmacy, 2022, doi:10.3390/pharmacy10030049_

Round 1
Reviewer 1 Report
Thank you for an interesting paper on how to improve uptake of vaccinations. Vaccine hesitancy is an important area to look at in light of the range of misinformation available to the public about Covid. There are a few areas that need clarification and addition.
Abstract. Please check your first sentence, it reads incomplete. Also, this should have a clear structure of background, methods, results and conclusion.
Introduction. For someone not familiar with Lean Six Sigma, please can you add in more information about what this is, evidence for use in this type of setting/research etc..
Materials and methods. Please can you justify why those particular practice sites were used. For this section, there is little information on recruitment. For example, how did you identify participants, how did you go by advertising and recruiting them, what was the procedure around arranging the interviews, was there an interview schedule (provide it in the text if there was), how many sites did you contact....?
Did you do content analysis or thematic analysis?
Any participant demographic data?
Table 1 is a bit unclear. Apologies if I missed something but was there supposed to be a number for each row in the Total column? How does this all add up to 990?
Table 2 should ideally be landscape format. There are also a few typos within the table. e.g uncontronalble, eleigible
Line 156 - type presumptive
Line 177 - typos recommendation, approach
Discussion and conclusion. Can you please comment on where you go from these findings. Any further plans to disseminate the guide or expand it to other vaccinations etc..
Line 245 - plethora of tools
Author Response
We kindly thank the reviewer for their time and insights in this review. We have provided a point by point response to each comment. We believe the paper has been strengthened by their input.
Thank you for an interesting paper on how to improve uptake of vaccinations. Vaccine hesitancy is an important area to look at in light of the range of misinformation available to the public about Covid. There are a few areas that need clarification and addition.
Abstract. Please check your first sentence, it reads incomplete. Also, this should have a clear structure of background, methods, results and conclusion.
- Thank you. We have addressed this introductory sentence. We have also followed formatting requirements from the journal in our abstract structure.
Introduction. For someone not familiar with Lean Six Sigma, please can you add in more information about what this is, evidence for use in this type of setting/research etc..
- Thank you for this comment. We agree with the author. This detail can be found a bit further on in the methods section.
Materials and methods. Please can you justify why those particular practice sites were used. For this section, there is little information on recruitment. For example, how did you identify participants, how did you go by advertising and recruiting them, what was the procedure around arranging the interviews, was there an interview schedule (provide it in the text if there was), how many sites did you contact....?
- Thank you, we have further clarified: "Sites were selected in collaboration with pharmacy chain leadership. All pharmacies in the Nashville (n=46) and Memphis (n=50) regions were included."
Did you do content analysis or thematic analysis?
- On page 3, line 99 we specify content analysis.
Any participant demographic data?
- We are unable to report per Walgreens agreement to protect confidentiality
Table 1 is a bit unclear. Apologies if I missed something but was there supposed to be a number for each row in the Total column? How does this all add up to 990?
- Thank you- this was an omission. Totals are now included.
Table 2 should ideally be landscape format. There are also a few typos within the table. e.g uncontronalble, eleigible
- Thank you. We have addressed spelling mistakes. We will work with editors on formatting as potentially an appendix.
Line 156 - type presumptive
- This appears correct upon review
Line 177 - typos recommendation, approach
- This appears correct upon review
Discussion and conclusion. Can you please comment on where you go from these findings. Any further plans to disseminate the guide or expand it to other vaccinations etc..
- Thank you. Beginning in line 256 we provide a description of future directions. We have also presented our work in citation #11 elsewhere.
Line 245 - plethora of tools
- Thank you
Reviewer 2 Report
Overall, the topic of research was very interesting. It discusses the use of LSS DMAIC strategy to improve vaccination recommendations in community pharmacies. I see the potential application of this strategy. However, there are certain sections that require improvement. Please check the attached file and address all of my comments. Great work.

Author Response
We kindly thank the reviewer for their time and insights in this review. We have provided a point by point response to each comment. We believe the paper has been strengthened by their input.
Overall, the topic of research was very interesting. It discusses the use of LSS DMAIC strategy to improve vaccination recommendations in community pharmacies. I see the potential application of this strategy. However, there are certain sections that require improvement. Please check the attached file and address all of my comments. Great work.
I feel the title is a bit lengthy and hard to understand. I would suggest to revise as follows: "The Lean Six Sigma Define, Measure, Analyze, Implement, Control (LSS DMAIC) Initiative: An Innovative Strategy for Quality Improvement of Pharmacist Vaccine Recommendations in Community Pharmacy"
- thank you, changed as recommended
This phrase is missing subject. So I would recommend to change to 'a highly accessible and convenient places for vaccination.'
- thank you, we have made this change.
Instead of using the word 'framework', change it to 'strategy'
- thank you for this comment, however lean six sigma is often referred to as a framework in the literature and so the authors after review have decided to maintain this language
Change it 'to help in protecting and significantly reducing'
- thank you, we have accepted this suggestion verbatim
change it to 'in evaluating vaccine hesitancy'.
- thank you, we have accepted this suggestion verbatim
Please remove extra space.
- thank you, we have done so
Just in curiosity, how was HIPAA maintained especially for phone conversation? While you have stated it was IRB exempt, I think it would be better to specify the efforts taken to ensure confidentiality during phone conversation, especially if interviews are taken by student research assistants. Also, was the supervisor or pharmacy preceptor available on-site for supervision?
- Thank you for the question. No patient information was abstracted for the purposes of the study. All data was provided de-identified to the study personnel.
This table is a bit hard to understand at a glance regarding how total score can be up to 990. One suggestion I would have is include range of minimum and maximum scores possible for each row. (e.g., 1 Process Step 1: 0-90, 3 Process Step 1: 0-60, etc.)
- Thank you, we have corrected this omission and completed the last column in the table
Okay, please make sure all contents in the same direction for certain columns. For example, SEV and question below are in different directions and hard to follow.
- Thank you. We will work with editors to format appropriately for the journal, including possibly adding as appendix to fix formatting issues.
One question. In the materials and methods section, you have listed several statistical tests. However, in this section, I only see descriptive analysis except for 24% difference in vaccination counts (and even for this sentence, did not specify the type of stat test used). Therefore, my suggestion is to clearly specify where each stat test was used and list in this section.
- Thank you for this comment. Generally, reporting of statistical test occurs within the methods rather than the results section. We have adhered to the journal's formatting requirements in this way. To your point specifically, this was a chi-square test that was used.
Why university logo has been cut? I would recommend to fully show everything or completely remove the university logo. Also, please enlarge overall figure. The contents are a bit hard to read.
- The university logo is included at the top of the figure. We have also enlarged the figure.
I would recommend to separate discussion and conclusion. Also, I would suggest to include limitation of this study.
- Thank you. We have added a conclusion and limitations section.